# The Effect of Heat Treatment on the Digestion and Absorption Properties of Protein in Sea Cucumber Body Wall

**DOI:** 10.3390/foods12152896

**Published:** 2023-07-29

**Authors:** Min Zhang, Yuxin Liu, Mengling Jin, Deyang Li, Ziye Wang, Pengfei Jiang, Dayong Zhou

**Affiliations:** 1School of Food Science and Technology, Dalian Polytechnic University, Dalian 116034, China; iszhangmin94@163.com (M.Z.); forever--xin@126.com (Y.L.); syjinmengling123@163.com (M.J.); dpuldy@163.com (D.L.); ziye76@163.com (Z.W.); 13840940070@163.com (P.J.); 2National Engineering Research Center of Seafood, Dalian 116034, China; 3Collaborative Innovation Center of Seafood Deep Processing, Dalian 116034, China

**Keywords:** sea cucumber body wall, heat treatment, oxidation, protein digestion, absorption, everted-rat-gut-sac model

## Abstract

This study was designed, for the first time, to investigate the effect of oxidation on the digestion and absorption properties of protein in boiled sea cucumber body wall (BSCBW) via simulated digestion combined with everted-rat-gut-sac models. Boiling heat treatments led to protein oxidation in SCBW, manifested by increases in free radical intensity, thiobarbituric acid reactive substances, carbonyl groups, disulfide bonds, dityrosine bonds, advanced glycation end products, protein hydrophobicity and aggregation, and declines in both free sulfhydryl groups and secondary structure transition from α-helix to β-sheet. Boiling for 2 h caused anti-digestion collagen unfolding, provided the action site for protease and improved protein digestion and absorption levels. On the contrary, excessive oxidative modification of 4 h BSCBW resulted in decreased protein digestion and absorption levels. From the perspective of texture, digestion and absorption properties, boiling for 2 h can obtain sea cucumber products with better edible and digestible properties, which is considered to be a better processing condition.

## 1. Introduction

Sea cucumber (*Stichopus japonicus*) is an important commercially developed variety with production reaching globally 22,270 tons in 2021 [1]. As the main edible part [2], consumers are keen on the sea cucumber body wall (SCBW) because of its special taste and high nutritional value. Fresh SCBW is rich in collagen and has a network structure maintained by collagen fibers and fibrillin microfibrils [3], which makes it very tough and hard to chew. Heat treatment can improve the texture and sensory properties of SCBW and give it excellent edible properties. In recent years, the health effects of food have attracted increasing attention from consumer. In addition to the sensory aspects of food, consumers tend to choose food based on nutritional quality. Studies have shown that heat treatment of aquatic products can affect the digestion properties and bioavailability of proteins during transit through the gastrointestinal tract, which has a serious impact on the nutritional value [4].

The digestibility and absorption of proteins in the small intestine are key factors that shape the nutritional quality of meat proteins. In general, most studies indicate that heat treatment affects the in vitro protein digestibility of meat in two parts. Short-time gentle heat treatment results in slightly higher levels of protein digestion, while severe heat treatment conditions (an increase in temperature or an extension of time) results in a decrease in in vitro protein digestibility [5]. This is because mild heat treatment induces protein denaturation and unfolding, thereby increasing the site of action for protease and thus improving digestibility [6]. However, interactions between hydrophobic regions within proteins occur as they unfold [4,7]. Meanwhile, heat-induced protein oxidation can lead to modifications of amino acids and the generation of various oxidation products, including carbonyl (from basic amino acids), disulfide bonds (from cysteine (thiol groups)) and dityrosine bonds (from tyrosine) [8]. The bonds between different proteins further originate the polymerization and aggregation of proteins [7,9]. Thus, access to the hydrolysis site becomes more difficult for proteases and protein hydrolysis is blocked, resulting in decreased digestibility [8]. However, so far the current research has mainly focused on the implications of heat treatment on the digestibility of meat, fish and seafood proteins [4], and few relevant studies have been conducted on collagen-dominated SCBW. 

Proteins are broken down by proteases in the digestive tract into free amino acids as well as smaller peptides (di-, tri- or tetrapeptides) then pass through the epithelium of the small intestine to enter the bloodstream by active transport or facilitative diffusion [10,11]. The digestibility of meat proteins determines the size and sequence of peptide segments and the composition of amino acids after digestion, which may affect their absorption [6,12]. Heat treatment can affect the digestibility of food proteins; hence, it can also affect the absorption of protein digestion products in the small intestine epithelial barrier. Current studies involve the effects of different heat treatments (high pressure, steam cooking) on the bioavailability of myofibril [12], myosin model systems [13] and oyster proteins [14]. However, their research objects were mainly limited to meat, protein model systems prepared from meat and aquatic muscle foods. Few studies have been reported on collagen-dominated sea cucumbers.

Our previous studies have shown that different boiling times had significant effects on the texture properties of SCBW. Insufficient heat treatment (100 °C for 0.5 h) caused substantial hardness and poor chewability of SCBW, while excessive heat treatment (100 °C for 4 h) led to soft, rotten and inelastic SCBW. Only moderate heat treatment (100 °C for 2 h) could give sea cucumber products moderate shear force, hardness and elasticity [15]. However, the protein digestion and absorption properties of sea cucumber with different degrees of heat treatment are not clear. Hence, this study aimed to study the effect of boiling time on protein digestion and absorption properties in boiled sea cucumber body wall (BSCBW) via simulated digestion combined with everted-rat-gut-sac models. Meanwhile, protein oxidation, crosslinking and aggregation in BSCBW with different treatment times were also investigated to reveal the mechanism behind the changes in protein digestion and absorption properties. 

## 2. Materials and Methods

### 2.1. Materials and Chemicals

Freshly harvested sea cucumbers (*Stichopus japonicus*) were chosen and purchased from the local aquatic market in Dalian, Liaoning, China with an average weight (100–120 g) and length (8–12 cm). The samples were always placed in ice sea water conditions during transport to the laboratory. All other reagents used in this work were of analytical grade and purchased from Kemiou Chemical Reagent Co., Ltd. (Tianjin, China).

### 2.2. Heat Treatment Program

Fresh sea cucumbers were divided into four groups after being dissected, gutted and washed. One group served as the control group without treatment, and the other three groups were heat treated at 100 °C for 0.5 h, 2 h and 4 h, respectively. They were named as fresh SCBW, 0.5 h boiled SCBW (0.5 h BSCBW), 2 h BSCBW and 4 h BSCBW. The samples were cooled and placed on ice for further analysis.

### 2.3. The Degree of Protein Oxidation

#### 2.3.1. Free Radical Intensity

The measurement of free radical intensity was implemented according to Liu’s method [16]. Freeze-dried sea cucumber powder (0.1 g) was placed in a 5 mm NMR tube (Wilmad, Buena, NJ, USA) and then placed in the sample cavity. Spectra was recorded via a Brucker 200 electron spin resonance (ESR) spectrometer (Karlsruhe, Germany) at room temperature. The free radical intensity was acquired through calculating the average value of the absolute values of the highest and lowest signal intensity on the ESR spectrum.

#### 2.3.2. Thiobarbituric Acid Reactive Substances Content (TBARS)

The measurement of TBARS was implemented according to Xie’s method [17]. Freeze-dried sea cucumber powder (0.5 g) was mixed with 2 mL of distilled water and 2 mL of 10% (*w*/*v*) trichloroacetic acid solution. After vortexation (2 min) and centrifugation (8000× *g*, 5 min), the supernatant was obtained. The supernatant (1 mL) and 0.01 M 2-thiobarbituric acid (TBA) solution (1 mL) were mixed and then heated in a boiling water bath for 25 min. The absorbance value of the mixture at 532 nm was determined. The TBARS value, expressed as mg malondialdehyde (MDA)/kg powder, was calculated from the standard curve prepared with 1,1,3,3-tetramethoxypropane as the MDA precursor.

#### 2.3.3. Protein Carbonyl Content

The measurement of the protein carbonyl content of SCBW was implemented by using a carbonyl assay kit (Jiancheng Technology Co., Nanjing, Jiangsu, China). Protein carbonyl reacts with 2,4-dinitrophenylhydrazide (DNPH) to produce 2,4-dinitrophenylhydrazone. The absorbance value of the product at 370 nm was determined. A molar extinction coefficient of 22,000 M^−1^ cm^−1^ was used for calculation. The carbonyl content was expressed as nmol/mg protein.

#### 2.3.4. Determination of Free Sulfhydryl Groups (SH) and Disulfide Bonds (S–S)

The measurement of SH (expressed as nmol/mg protein) and S–S (expressed as nmol/mg protein) of SCBW were implemented according to Ellman’s method [18]. The samples were dissolved in a urea SDS solution (8.0 M urea, 3% SDS, 0.1 M phosphate, pH 7.4) and incubated with 5,5-dithiobis (2-nitrobenzoic acid) (DTNB) reagent at 25 °C for 15 min. The absorbance value at 412 nm was determined. Reagent blanks and sample blanks were run simultaneously. A molar extinction coefficient of 13,600 M^−1^ cm^−1^ was used for calculation.

#### 2.3.5. Structural Changes of Water-Soluble Fraction

Boiled sea cucumbers and cooking liquors were homogenized together, and then were centrifuged at 13,500× *g* for 20 min (40 °C). The supernatant obtained was taken as the water-soluble fraction. For fresh samples, the operation process was identical to the boiled samples, with the exception that the operation temperature was maintained at 4 °C.

The water-soluble fraction obtained above was diluted to 0.2 mg/mL with distilled water. Then, dityrosine intensity [19], fluorescence spectroscopy [20] and advanced glycation end product (AGE) intensity [21] were measured by using a Hitachi F-2700 spectrofluorimeter (Hitachi, Tokyo, Japan). The widths of both the excitation slit and emission slit were set as 10.0 nm. For dityrosine intensity, the excitation wavelength was 325 nm and the emission wavelength was 420 nm. For fluorescence measurements, the excitation wavelength was 280 nm, and the scanning range of the emission wavelength was 300–500 nm. The scan speed was 1500 nm/min and the experimental temperature was 25 °C. For AGE intensity, the excitation and emission wavelengths were set to 370 nm and 440 nm, respectively.

The water-soluble fraction obtained above was diluted to 0.025 mg/mL with distilled water. Circular dichroism (CD) spectra [20] were measured in the extreme ultraviolet region of 190–250 nm at 25 °C using a Jasco J-1500 CD spectropolarimeter (Jasco Corporation, Tokyo, Japan). The measurements were repeated three times to obtain CD images. The fractions of α-helix, β-sheet, β-turn, and random coil were obtained using the Secondary Structure Estimation software (Spectra manager V2.01.01, Jasco Corporation, Tokyo, Japan).

#### 2.3.6. Protein Surface Hydrophobicity

The measurement of sea cucumber protein hydrophobicity was implemented according to Chelh’s method [22]. Briefly, the water-soluble fraction obtained above was diluted to 1 mg/mL with distilled water. Firstly, the protein diluent and bromophenol blue solution were mixed. The absorbance value of supernatant at 595 nm was determined after incubation (10 min), centrifugation (7500 rpm, 15 min, 4 °C) and dilution (10 times). The control group received distilled water instead of a protein solution. The hydrophobicity of sea cucumber protein was expressed as the amount of bromophenol blue binding (BPB).
$$BPB(\mu g)=\frac{A1-A2}{A1}\times 200$$
where A1 represents the absorbance value of the control group and A2 represents the absorbance value of the sample.

#### 2.3.7. Protein Aggregation

The measurement of sea cucumber protein aggregation was implemented according to Santé-Lhoutellier’s method [8]. Briefly, the water-soluble fraction obtained above was diluted to 1 mg/mL with distilled water. The fluorescence intensity of the mixture sample of protein diluent and Nile Red solution was measured by using a Hitachi F-2700 spectrofluorimeter (Hitachi, Japan). The excitation wavelength was 560 nm, the emission wavelength was 620 nm, the excitation slit width was 10.0 nm and the emission slit width was 10.0 nm.

### 2.4. In Vitro Simulated Gastrointestinal Tract Digestion

The static digestion of sea cucumber protein was simulated by gastrointestinal digestion in vitro according to Minekus’s method [23]. Sea cucumber freeze-dried powder (0.05 g) was vorticized with 6 mL of simulated salivary fluid (SSF) and then the mixture was incubated by shaking at 100 rpm in a 37 °C water bath for 10 min. Then 6 mL of simulated gastric fluid (SGF) was added and the digestive system was adjusted to pH 3.0. The mixture was further incubated for 120 min. After the simulated gastric digestion, 12 mL of simulated intestinal fluid (SIF) was added to the chyme, and the mixture was adjusted to pH 7.0 and further incubated for 120 min. Then the solution was boiled for 10 min to stop the digestion reaction. This made the digestion samples suitable for subsequent experiments.

#### 2.4.1. Release of the Primary Free Amino Group

The primary free amino group of the digestion sample was quantified using the OPA method [24]. The supernatant was obtained after the digestion sample was centrifuged at 12,000× *g* for 10 min, which was the sample solution. After taking 200 μL of the sample solution, 1.5 mL of OPA reagent was added and the mixture was incubated for 2 min, and the absorbance value was immediately determined at 340 nm. The time of recording the absorbance value must be synchronized because the absorbance value changes with time. At the same time, the concentration curve of serine standard solution (0.1 mg/mL) was drawn, and the free amino group content was calculated according to the standard curve.

#### 2.4.2. Determination of Trichloroacetic Acid: Soluble Peptide Yield (TCA-Ysp)

The TCA-Ysp was measured according to the method reported by Chen [25] with some modifications. The digestion sample (500 μL) was mixed with 20% (*w*/*w*) TCA (500 μL). Then the mixture was centrifugated at 12,000 rpm for 10 min (4 °C). The peptide content in the supernatant was determined by the Folin-phenol method. After mixing the supernatant (100 μL) and Folin-phenol A solution (500 μL), the mixture was incubated at room temperature for 10 min. Folin-phenol B solution (50 μL) was added, mixed and incubated in 30 °C water bath for 30 min, and cooled naturally. The absorbance (500 nm) was measured. Meanwhile, the standard curve of BSA as the standard protein was drawn. TCA-Ysp (%) was obtained by calculating the ratio of peptide content in the supernatant to total protein before digestion.

### 2.5. Everted Gut Sac Model

#### 2.5.1. Construction of Everted-Rat-Gut Sacs

Male Sprague–Dawley rats (100–150 g) were chosen and purchased from Liaoning Changsheng Biotechnology Co., Ltd. (Benxi, Liaoning, China). All animal procedures were subjected to the approval of the Committee on the Ethics of Animal Experiments of Dalian Polytechnic University (approval number: DLPU2022037). All procedures for animal experiments followed the guidelines of the National Institutes of Health. The constructed everted-rat-gut sacs were obtained according to Yin’s method [26].

#### 2.5.2. Digestion Product Pass across the Rat Gut Wall

The supernatant was obtained after the digestion sample was centrifuged at 12,000× *g* for 10 min, which was the sample solution. The latter was added to the mucosal side, i.e., 7 mL sample solution was lyophilized and redissolved in 7 mL Krebs–Ringer buffer solution.

After incubation at 37 °C for 0, 20, 40, 60, 80, 100 and 120 min, 50 μL of serosal fluid was taken out and mixed with 50 μL of methanol to terminate the reaction. At the same time, serosal fluid was filled with 50 μL of Krebs–Ringer buffer. The serosal fluid sample supernatant was collected after centrifugation (19,000× *g*, 4 °C, 25 min) to determine the peptide content. 

#### 2.5.3. Determination of Total Amino Acid Absorption Rate

The measurement of amino acid content was carried out via an automatic amino acid analyzer (LA8080, Hitachi, Tokyo, Japan) [27]. The total amino acid absorption rate was obtained by calculating the ratio of the content of total amino acid absorbed to the serosal side (inside the sac) to that of the simulated gastrointestinal digestion product.

#### 2.5.4. Determination of Peptide Absorption Rate

The peptide absorption of digestion products from 0 min to 120 min was measured by the Folin-phenol method. The peptide absorption rate was evaluated by calculating the ratio of the content of peptide absorbed to the serosal side (inside the sac) to that of the simulated gastrointestinal digestion product.
$$Peptide\;absorption\;rate\;(\%)=\frac{\left(Ci\times Vi)-(Cb\times Vb\right)}{CV\times A}\times 100$$
where Ci is the peptide content of the serosal side of sample (mg/mL), Vi is the volume of solution of the serosal side of sample (mL), Cb is the peptide content of the serosal side of the blank (mg/mL), Vb is the volume of solution of the serosal side of the blank (mL), C is the initial peptide content of the mucosal side (outside the sac) (mg/mL), V is the initial volume of solution of the mucosal side (mL) and A is surface area of the small intestinal (cm^2^).

#### 2.5.5. Determination of Peptide Content by HPLC-UV Analysis

HPLC (Shimadzu LC-20AD, Tokyo, Japan) was used to analyze the peptide content coupled with an Elite C18 analytical column (4.6 × 250 mm, 5 μm). Mobile phase: A was 10% methanol; B was methanol. Gradient elution steps: 0–15 min, 25% B; 15–25 min, 25–90% B; 25–26 min, 95–25% B; 26–35 min, 25% B. Injection volume: 10 μL; flow rate: 0.5 mL/min; column temperature: 25 °C; monitored wavelength of the chromatogram: 214 nm.

### 2.6. Statistical Analysis

The everted gut sac experiment was carried out five times in parallel, the other experiments were carried out three times in parallel and the values are shown as means ± standard deviation. Significant differences were measured via one-way ANOVA and Tukey’s test by using SPSS 22.0 software (SPSS Inc., Chicago, IL, USA). *p* < 0.05 means the difference was significant.

## 3. Results and Discussion 

### 3.1. Changes in Oxidative Indicators

With the extension of boiling time, the free radical intensity (Figure 1A,B), TBARS (Figure 1C), carbonyl content (Figure 1D), disulfide bond content (Figure 1F), dityrosine bond content (Figure 1G), fluorescence intensity (Figure 1H) and AGE content (Figure 1I) of SCBW increased, while the free sulfhydryl group content (Figure 1E) decreased. The results showed that protein and lipid oxidation occurred. Moreover, the oxidation degree was enhanced with the extension of heat treatment time. Proteins are generally oxidized and generate free radicals and TBARS during heat treatment. The increase in free radical production can significantly lead to the modification of the amino acid side chain and protein backbone, manifested in the formation of carbonyls (from basic amino acids), disulfide bonds (from cysteine (thiol groups)) and dityrosine bonds (from tyrosine), as well as the reduction in thiol groups and tryptophan [8]. The heat treatment of other aquatic animal foods, such as abalone [28] and scallops [29], has been observed to result in intensified protein oxidation as the duration of heat treatment increases.

### 3.2. Structural Modifications of Proteins

Circular dichroism spectra have been widely used in elucidating structural changes of proteins/peptides at the secondary folding level under heating treatment [30]. The changes of secondary structure in fresh and boiled SCBW with different boiling times are summarized in Table 1. The levels of α-helix, β-sheet, β-turn and random coil of fresh SCBW were 37.50 ± 3.25, 18.87 ± 6.30, 15.53 ± 0.76 and 28.13 ± 2.44%, respectively. During boiling, α-helix and β-turn fractions decreased significantly, accompanied by a rise in β-sheet and random coil fractions, indicating a structural shift from an ordered to a disordered state of the protein, which was consistent with the report of Yan et al. [31]. This is possibly due to the exposure of hydrophobic regions and the breakdown of hydrogen bonds. On the one hand, heat treatment possibly accelerates the denaturation and folding process of protein by loosening the protein structure. On the other hand, long-term high-temperature treatment may lead to the enhancement of the kinetic energy of protein molecules and vibration of polar groups, thus changing the secondary structures [32].

The surface hydrophobicity of proteins can be used to evaluate the conformational change of proteins [8]. The protein surface hydrophobicity increased observably from 82.62 ± 1.38 μg Bound BPB (fresh SCBW) to 93.68 ± 0.33 μg Bound BPB (boiled for 4 h) with the extension of boiling time (Figure 2A). Hydrophobic amino acids in their natural state are buried in the protein folding core. Heat treatment facilitates the unfolding of the protein structure and exposes the amino acids to the protein surface [10].

The specificity of fluorescence emitted by the combination of Nile red with aggregates was used to evaluate the degree of protein aggregation [8]. Compared to fresh SCBW (438.90 ± 0.30), the fluorescence intensity significantly increased to 737.30 ± 0.26 (boiled for 4 h) with the extension of boiling time (Figure 2B). This is mainly attributed to the formation of high-molecular-weight aggregates derived from the disruption of protein conformation and the release of hydrophobic groups during heat treatment [33]. The aforementioned finding was also documented in the investigation conducted by Li et al. [33]. That is, protein oxidation caused the formation of crosslinks between cysteine (disulfide bonds) or tyrosine (dityrosine bonds) [10] and the alteration of secondary and tertiary structures. This further leads to an increase in the unfolding state of the protein structure and protein–protein interaction, resulting in protein aggregation at last.

### 3.3. Protein Digestion Properties of Boiled SCBW

During gastrointestinal tract (GIT) digestion, the protein hydrolysis degree can be evaluated by measuring the release of primary free amino groups and the formation of peptides [34]. So, the OPA method was used to quantify the free amino group in gastric and intestinal digestion products to assess the protein digestion level, as shown in Figure 3A. Further, the TCA precipitation method combined with the Folin-phenol method was used to quantify peptides (<10 amino acids) and free amino acids generated during digestion to confirm the progress of hydrolysis [35]; the result are demonstrated in Figure 3B. The release of free amino groups in small intestinal digestion was significantly increased in comparison with gastric digestion, indicating that protein was mainly digested in the intestine (Figure 3A). During gastric and gastrointestinal digestion, the content of free amino groups released by protein hydrolysis (Figure 3A) and TCA-soluble peptide yield (Ysp) (Figure 3B) intensified at first and then dropped with the prolongation of boiling time. The levels of free amino groups and Ysp released in 2 h BSCBW were the highest, and those in 4 h BSCBW decreased. 

During digestion, the peptide bonds of proteins are hydrolyzed by proteases and free amino groups are released [36]. Type I collagen in fresh SCBW has strong resistance to pepsin due to its tight spatial structure [37], thus a lower degree of hydrolysis was shown. Moderate heating (from 0.5 h to 2 h) induced the partial expansion or oxidation of type I collagen, exposing more active sites for pepsin action (hydrophobic residues including Phe, Tyr, Trp and Leu) and improving the hydrolysis degree [38]. Conversely, excessive heating (4 h) led to an increased degree of oxidation of proteins, which were more prone to cross-linking and aggregation, which enhanced resistance to enzymolytic proteins and reduced the hydrolysis degree [8]. However, during the gastrointestinal digestion stage, protein hydrolysis massively increased under trypsin action, which led to structural collapse, which could rule out the theory that oxidation-induced molecular aggregation is the reason that protease is prevented from approaching the cleavage site. Thus, the decrease in hydrolysis degree was attributed to modification of multiple amino acid residues or enzyme cleavage sites, resulting in digestive enzymes failing to recognize the sites correctly. Kaur et al. [38] found that long cooking of beef produced limited peptides, which were not further decomposed into free amino acids by digestive enzymes. This could be attributed to the modification of various amino acid residues, for instance, the oxidative modification of aromatic amino acid residues (pepsin hydrolysis site). In this study, the change of Ysp was consistent with that of the release of free amino groups, indicating that boiling for 2 h can raise the digestion level of protein, while boiling for 4 h can reduce the digestion level.

### 3.4. Absorption Properties Determined by the Everted-Rat-Gut-Sac Model 

The degree to which food compounds are absorbed by intestinal epithelial cells and enter the systemic circulation after digestion in the gastrointestinal tract is considered as bioavailability, which is related to the stability of the compounds after digestion in the gastrointestinal tract and the absorption efficiency through intestinal epithelial cells [39]. In the model, in a process that simulates GIT digestion, food proteins are further denatured and hydrolyzed by pepsin to form a mixture of polypeptides and free amino acids. Amino acids are not absorbed in this process. Next, the digested mixture is further hydrolyzed by trypsin into free amino acids, dipeptides and tripeptides, which are further absorbed through the small intestinal lumen via several transport modes [40]. Peptidases in the small intestine barrier also further hydrolyze peptides and affect their biological activity. Therefore, an in vitro tissue-based model of everted sac gut was used to determine peptide absorption in this study. In addition, total amino acid absorption rate (Figure 4A) and absorption properties of peptides (Figure 4B) were measured to evaluate intestinal absorption.

In the constructed everted sac model, amino acids undergo an absorption process from the mucosal side (outside the sac) to the serosal side (inside the sac). The absorption rate of total amino acids on the serosal side was calculated from the ratio of amino acids inside the sac to the amino acids outside the sac, as demonstrated in Figure 4A. The peptide absorption of digestion products from 0 min to 120 min was measured using the Folin-phenol method, which was evaluated by calculating the ratio of peptide content of absorbed to the serosal side (inside the sac) to that of the simulated gastrointestinal digestion product, as shown in Figure 4B. Further, HPLC was used to determine peptide absorption of SCBW with different boiling times in the everted sac model (Figure 5 and Figure 6).

With the prolongation of boiling time, the total amino acid absorption rate of protein digestion products of sea cucumber firstly intensified and then dropped. Moreover, the digestion products of 2 h BSCBW showed the highest total amino acid absorption rate (Figure 4A). With the progress of incubation time, the peptide absorption rate (Figure 4B) and the peak area (Figure 6A–D) of the three major peptide peaks of the chromatogram of serosal side samples (Figure 5A–D) gradually increased. This suggested that the level of peptides absorbed from mucosal side to the serosal side was raised with the passing of time. The heat treatments on SCBW improved the peptides absorption. However, 4 h heat treatments obviously dropped the peptide absorption level (Figure 6E).

### 3.5. Heatmap and Correlation Analysis

Association between the evaluation indexes of protein oxidation (free radical intensity, TBARS, carbonyl groups, S–H, S–S, dityrosine bonds, AGEs, Ho and protein aggregation), gastrointestinal tract digestion levels (DH and Ysp) and absorption levels (amino acid and peptide absorption) of SCBW with different boiling times were analyzed intuitively in a heat map, as shown in Figure 7A. During boiling, the oxidation degree of sea cucumber protein showed a time-dependent increase. The levels of gastrointestinal tract digestion and absorption of sea cucumber protein intensified first and then dropped, and the best digestion and absorption properties were observed in 2 h BSCBW.

Obviously, there was an inflection point in the change of digestion and absorption properties in 2 h BSCBW. Therefore, sea cucumber samples before and after boiling for 2 h were selected, respectively, to conduct a correlation analysis between their oxidative indicators and protein digestion and absorption indicators; the results are demonstrated in Figure 7B,C. The gastrointestinal tract digestion levels and absorption levels of protein in fresh SCBW, 0.5 h BSCBW and 2 h BSCBW were positively correlated with oxidation degree (Figure 7B). However, protein digestion and absorption levels in 2 h BSCBW and 4 h BSCBW were negatively correlated with protein oxidation levels (Figure 7C). In addition, protein digestion and absorption showed an obvious positive correlation. Therefore, it could be considered that the oxidation degree of 2 h BSCBW was appropriate and both promoted protein hydrolysis and improved the absorption level of peptides and amino acids. On the contrary, excessive oxidative modification of 4 h BSCBW resulted in the inability of digestive enzymes to recognize sites, affecting protein hydrolysis and reducing absorption levels.

In general, the heat procedures of aquatic muscle foods (such as shrimp, oysters, scallop, abalone, etc.) include steaming, boiling, roasting, frying and baking. The above aquatic muscle foods mainly contain myofibrillar proteins, and their digestibility tends to decline after heat treatment. Microwave treatment (75–125 °C, 5–15 min) showed a negative effect on the peptide production and protein digestibility of shrimp when compared with untreated samples [32]. Steaming (100 °C, 10–15 min), baking (250 °C, 15–25 min) and firing (160 °C, 1.5–3 min) treatment harmed significantly the in vitro digestibility and mineral bioaccessibility of raw oysters [41]. Sea cucumber is rich in collagen, and its heat treatment and digestion properties are completely different from those of the aquatic muscle food mentioned above. Since type I collagen is an anti-digestion protein [37], the in vitro digestibility of fresh sea cucumber protein is much lower than that of aquatic muscle food. In addition, our previous study [15] found that the hierarchy of collagen becomes loose after boiling for 0.5 h, which was manifested as collagen fiber depolymerization and collagen fibril unfolding. After boiling for 2 h, collagen fibrils broke down into collagen microfibrils. In addition, under the joint action of collagen gelatinization, a spongy network structure was formed. Therefore, it can be speculated that 2 h heat treatment increased the action site of protease and was conducive to improving the degree of protein hydrolysis. However, excessive boiling (4 h) led to a collapse of the gelatin network. This may belong to another type of aggregation phenomenon, resulting in the covering of a large amount of protease action sites, reducing the degree of hydrolysis. On the other hand, the lower digestibility of proteins means fewer absorbed nutrients. Therefore, in this study, the protein digestion and absorption properties of sea cucumber protein first increased and then dropped with the extension of heat treatment time. Furthermore, the best digestion and absorption properties were observed in 2 h BSCBW.

## 4. Conclusions

Boiling at 100 °C led to protein oxidation of SCBW, which was shown as increased free radical intensity, TBARS, carbonyl groups, disulfide bonds, dityrosine bonds, AGEs, protein hydrophobicity and aggregation, as well as decreases in free sulfhydryl groups. Being different from aquatic muscle food, collagen-rich sea cucumber still maintained an excellent spongy porous structure after boiling for 2 h at 100 °C. This may be due to the protein structure unfolding caused by moderate oxidation, providing action sites for protease and improving protein digestion and absorption properties. In contrast, excessive oxidative modification of 4 h BSCBW resulted in the inability of digestion enzymes to recognize sites, affecting protein hydrolysis and reducing absorption levels. Therefore, no matter whether from the perspective of texture or digestion and absorption properties, boiling for 2 h at 100 °C can obtain sea cucumber products with better edible and digestible properties, which is considered to be a better processing condition. However, from the perspective of protein molecules, the mechanism of the effect of boiling on the digestibility of sea cucumber protein is still unclear, which needs to be addressed in future research.

## Figures and Tables

**Figure 1 foods-12-02896-f001:**
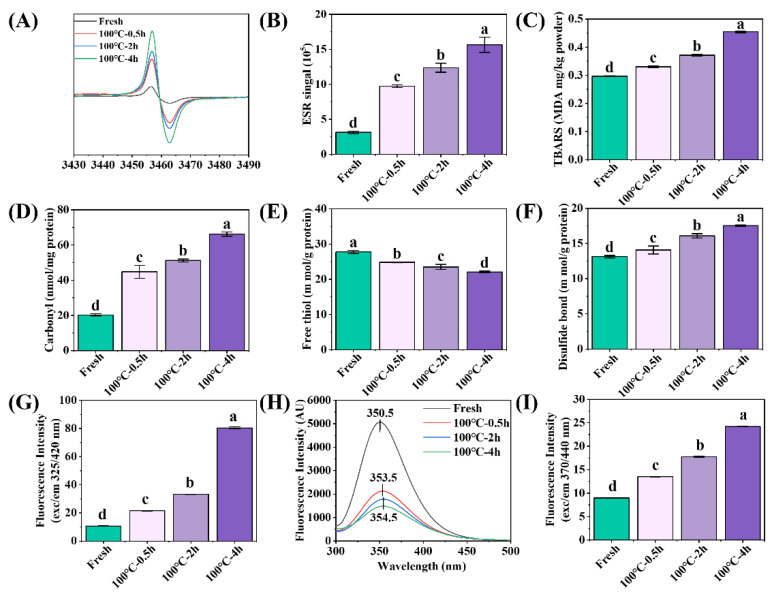
Changes in oxidative indicators of sea cucumber body wall (SCBW) with different treatments. (**A**) electron spin resonance (ESR) spectrum; (**B**) free radical intensity; (**C**) thiobarbituric acid reactive substances (TBARS); (**D**) carbonyl content; (**E**) free sulfhydryl group content; (**F**) disulfide bonds; (**G**) dityrosine bonds; (**H**) fluorescence intensity and (**I**) advanced glycation end products (AGEs). Values of different groups with different lowercase letters differ significantly (*p* < 0.05).

**Figure 2 foods-12-02896-f002:**
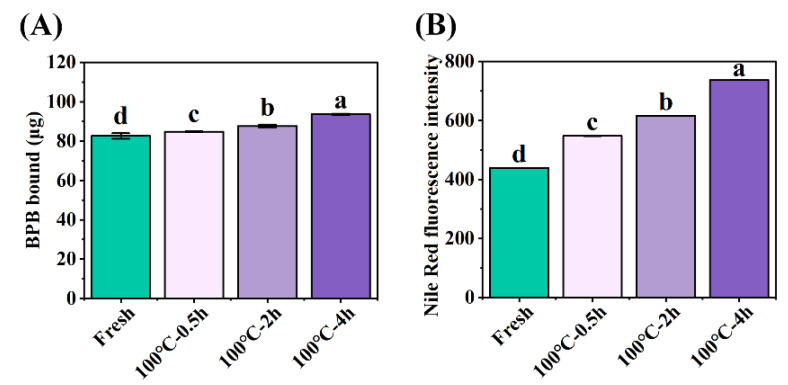
Changes in (**A**) protein surface hydrophobicity and (**B**) protein aggregation of SCBW with different treatments. Values of different groups with different lowercase letters differ significantly (*p* < 0.05).

**Figure 3 foods-12-02896-f003:**
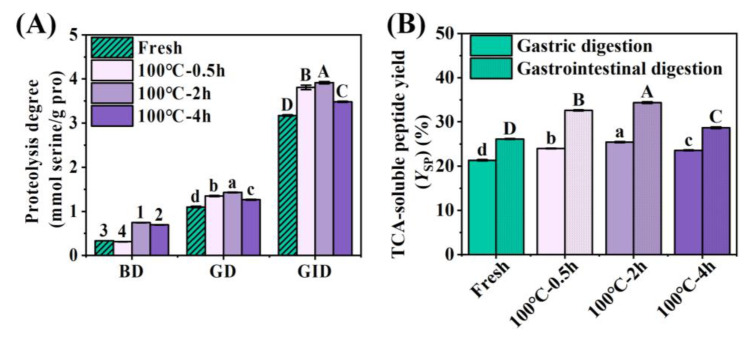
Changes in gastric and gastrointestinal digestion of SCBW with different treatments. (**A**) proteolysis degree (BD, before digestion; GD, after gastric digestion; GID, after gastrointestinal digestion); (**B**) TCA-soluble peptide yield. Values of different groups with different letters differ significantly (*p* < 0.05). Numbers (1–4) indicated significant differences between treatment group samples before digestion, lower letters (a–d) indicated significant differences between treatment group samples after gastric digestion, and upper letters (A–D) indicated significant differences between treatment group samples after gastrointestinal digestion.

**Figure 4 foods-12-02896-f004:**
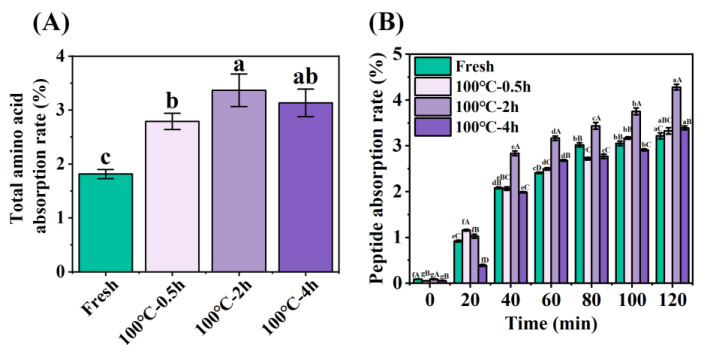
Changes in (**A**) total amino acid absorption rate and (**B**) peptide absorption rate across everted-rat-gut sacs of SCBW with different treatments. Values of different groups with different letters differ significantly (*p* < 0.05). In Figure 4A, lower letters (a–d) indicated significant differences between different treatment groups (Fresh, 100 °C-0.5 h, 100 °C-2 h, 100 °C-4 h) after absorption for 120 min. Upper letters (A–D) in Figure 4B indicated significant differences between the same absorption time of different treatment groups. Lower letters (a–g) indicated significant differences between different absorption times within the same treatment group.

**Figure 5 foods-12-02896-f005:**
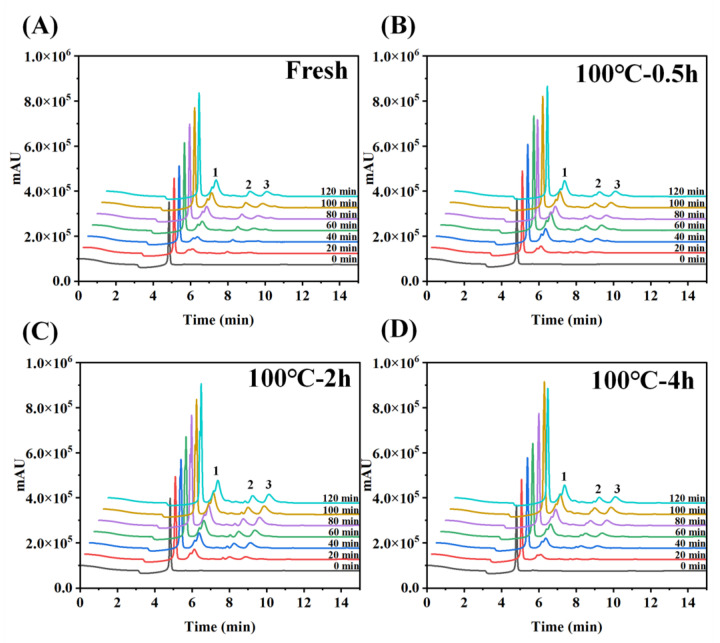
The protein absorption properties of sea cucumber digestion products with different treatment groups. (**A**–**D**) represent the liquid chromatogram of serosal fluid samples of digestion product incubated after 0, 20, 40, 60, 80, 100 and, 120 min incubation. Peaks 1–3 correspond to polypeptides absorbed by sea cucumber digestion products through the everted-rat-gut-sac.

**Figure 6 foods-12-02896-f006:**
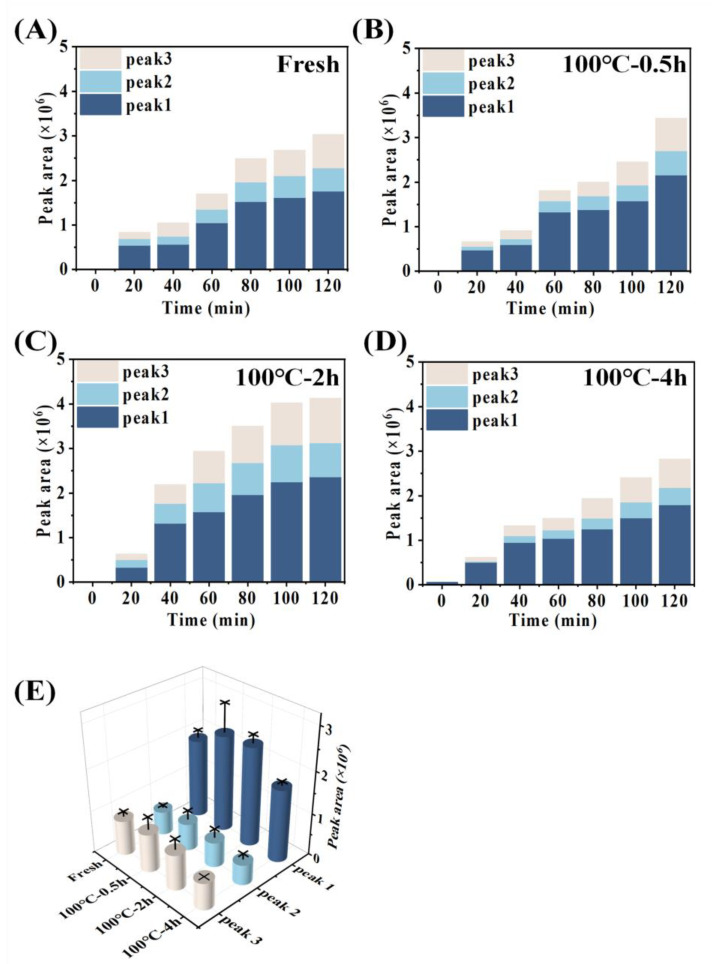
Changes in the polypeptide content transported across the everted-rat-gut sacs. The peak area of serosal fluids incubated with (**A**) fresh SCBW, (**B**) 0.5 h BSCBW, (**C**) 2 h BSCBW and (**D**) 4 h BSCBW digestion products. (**E**) The peak area of serosal fluids after 120 min incubation with different sea cucumber digestion products.

**Figure 7 foods-12-02896-f007:**
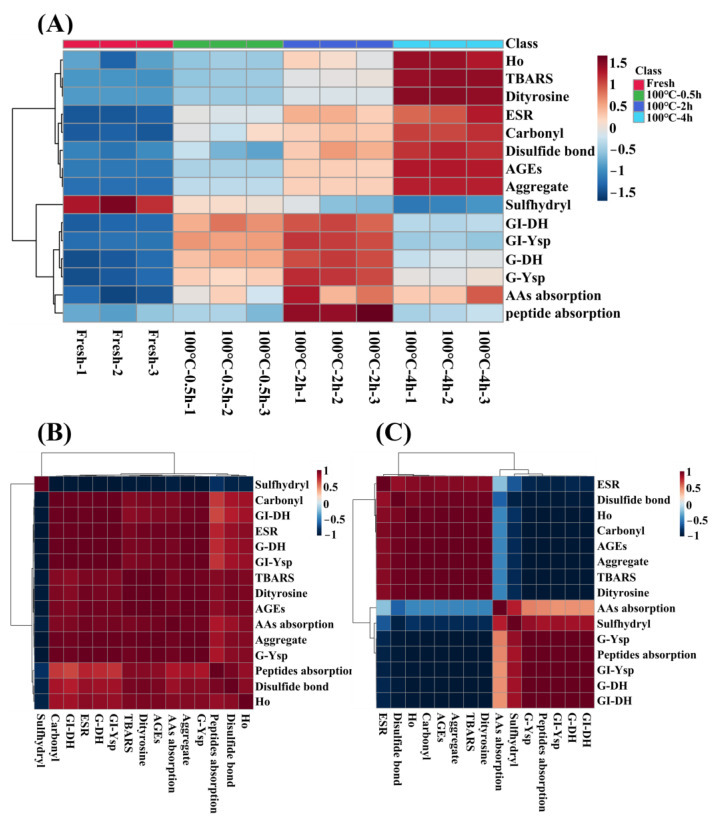
The heatmap analysis and correlation analysis of protein oxidative indicators (free radicals, TBARS, carbonyl groups, sulfhydryl groups, disulfide bonds, dityrosine, AGEs, protein hydrophobicity (Ho) and aggregation); protein digestion properties (gastric degree of hydrolysis (G-DH) and gastrointestinal DH (GI-DH) and TCA-soluble peptide yield of gastric (G-Ysp), gastrointestinal (GI-Ysp)) and absorption properties (amino acid and peptide absorption) of SCBW with different heat treatment. (**A**) the heatmap analysis of SCBWs with different treatments; (**B**) the correlation analysis between fresh SCBW, 0.5 h-BSCBW and 2 h-BSCBW; (**C**) the correlation analysis between 2 h BSCBW and 4 h BSCBW.

**Table 1 foods-12-02896-t001:** Changes in the secondary structure of protein in SCBWs with different treatments.

%	Fresh	100 °C for 0.5 h	100 °C for 2 h	100 °C for 4 h
α-helix	37.50 ± 3.25 a	28.10 ± 5.52 b	7.33 ± 1.11 c	0.27 ± 0.25 d
β-sheet	18.87 ± 6.30 c	39.40 ± 8.79 b	52.83 ± 1.11 c	55.17 ± 0.80 a
β-turn	15.53 ± 0.76 a	4.23 ± 2.37 b	3.07 ± 2.45 b	3.13 ± 0.72 c
random coil	28.13 ± 2.44 a	28.27 ± 2.80 a	36.80 ± 2.17 b	41.7 ± 1.37 b

Different letters (a–d) in the same line differ significantly (*p* < 0.05).

## Data Availability

The data used to support the findings of this study can be made available by the corresponding author upon request.

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
