# Peer review of "The Effect of Heat Treatment on the Digestion and Absorption Properties of Protein in Sea Cucumber Body Wall"

_foods, 2023, doi:10.3390/foods12152896_

Round 1
Reviewer 1 Report
Thank you very much for article submission in this journal. Paper explained the Effect of heat treatment on digestion and transport properties 2 of protein in sea cucumber body wall. Article is written good and topic of intrest. However there are some questions need to be addressed.
1. Add more recent references related to study in introduction.
2. Also check reference style. Must follow journal guidelines.
3. Details should be given in material and methods section.
English editing can be done
Reviewer 2 Report
Reviewer Report:
This paper aims to investigate the effect of oxidation on protein digestion and transport properties in boiled sea cucumber body wall (BSCBW) via simulated digestion combine with everted-rat-gut-sac models.
In this study, authors showed that the boiling at 100 °C led to protein oxidation of SCBW, which is different from aquatic muscle food, collagen-rich sea cucumber still maintained an excellent spongy porous structure after boiling for 2 h at 100 °C.
The manuscript is written comprehensively enough to be understandable; the paper stated the purpose, discussion and global implication are clearly stated and consistent with the rest of the manuscript; authors provided the required tests and analysis.
The authors addressed their hypothesis and opinion in a reproducible way and proved their results through all the required experiments and analysis. The results were presented in a clear way which facilitate in reaching a conclusion elucidates that no matter from the perspective of texture or digestion and transport properties, boiling for 2 h at 100 °C can obtain sea cucumber products with better edible and digestible properties, which is considered to be a better processing condition.
Authors did not use enough number of references to prove their results and to enrich their introduction and discussion. I suggest adding more references about the:
Sea cucumber (Stichopus japonicus)
Effect of temperature or an extension of time on protein digestibility
Hydrophobic regions within proteins/protease/ proteolysis.
The abbreviations were explained at the first place they are mentioned.
In vitro, in vivo, et al, via.: should be written in italic.
No plagiarism has been detected.
References: The authors did not follow the journal guidelines for some references.
Year: should be in Bold, Volume: should be in italic (reference 3: add the volume)
Reviewer 3 Report

Can be improved
Round 2
Reviewer 3 Report
The manuscript is improved and can be published